# SASSL: Enhancing Self-Supervised Learning via Neural Style Transfer

## Abstract

Self-supervised learning relies heavily on data augmentation to extract meaningful representations from unlabeled images. While existing state-of-the-art augmentation pipelines incorporate a wide range of primitive transformations, these often disregard natural image structure. Thus, augmented samples can exhibit degraded semantic information and low stylistic diversity, affecting downstream performance of self-supervised representations. To overcome this, we propose *SASSL: Style Augmentations for Self Supervised Learning*, a novel augmentation technique based on Neural Style Transfer. The method decouples semantic and stylistic attributes in images and applies transformations exclusively to the style while preserving content, generating diverse augmented samples that better retain their semantic properties. Experimental results show our technique achieves a top-1 classification performance improvement of more than 2% on ImageNet compared to the well-established MoCo v2. We also measure transfer learning performance across five diverse datasets, observing significant improvements of up to 3.75%. Our experiments indicate that decoupling style from content information and transferring style across datasets to diversify augmentations can significantly improve downstream performance of self-supervised representations.

## 1 Introduction

Data labelling is a challenging and expensive process, which often serves as a barrier to build machine learning models to solve real-world problems. Self-supervised learning (SSL) is an emerging machine learning paradigm that helps to alleviate the challenges of data labelling, by using large corpora of unlabeled data to pre-train models to learn robust and general representations. These representations can be efficiently transferred to downstream tasks, resulting in performant models which can be constructed without access to large pools of labeled data. SSL methods have shown promising results in recent years, matching and in some cases exceeding the performance of bespoke supervised models with small amounts of labelled data.

Given the lack of labels, SSL relies on pretext tasks, *i.e.*, predefined tasks where pseudo-labels can be generated. Some examples include contrastive learning Chen et al. (2020a); He et al. (2020), clustering Caron et al. (2021; 2020); Assran et al. (2022), and generative modeling He et al. (2022); Devlin et al. (2018). Many of these pretext tasks involve training the model to distinguish between different views of the same input and inputs corresponding to different samples. For these tasks, the way input data is augmented is crucial for the network to learn useful invariances and extract robust representations (Chen et al., 2020a). While state-of-the-art augmentations incorporate a wide range of primitive color, spectral and spatial transformations, they often disregard the natural structure of an image. As a result, SSL pre-training methods may generate augmented samples with degraded semantic information, and may be less able to capture diverse visual attributes.

To overcome this limitation, we propose *Style Augmentations for Self Supervised Learning (SASSL)*, a novel SSL data augmentation technique based on Neural Style Transfer to obtain stronger augmented samples via semantically-aware preprocessing. In contrast to handcrafted augmentation techniques operating on specific formats (e.g. pixel or spectral domain), SASSL disentangles an image into perceptual (style) and semantic (content) representations that are learned from data. By applying transformations exclusively to the style of an image while preserving its content, we can generate augmented samples with diverse styles that retain the original semantic properties.

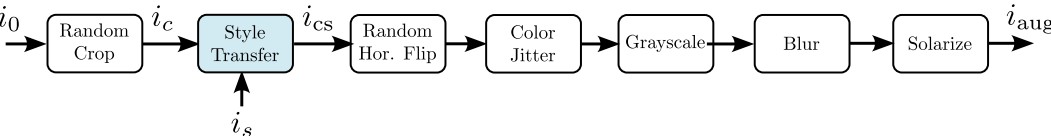

(a) **Proposed data augmentation pipeline**.

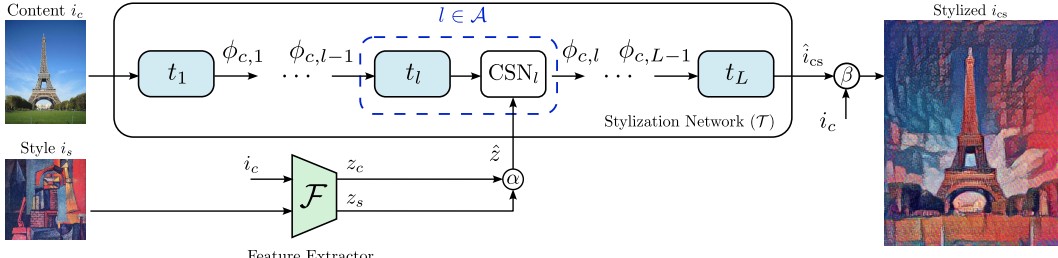

(b) **Style transfer preprocessing**.

Figure 1: **Towards diverse SSL data augmentation via neural style transfer.** We propose *SASSL*, a novel augmentation method based on style transfer that generates semantic-aware transformations of pre-training images by operating exclusively on their style. Given a pre-training image and an external style representation, SASSL combines the content attributes of the image with the external style, producing an augmented view that better preserves semantic attributes. By controlling the stylization strength via feature and pixel interpolations, we empirically show SASSL promotes robust representations by incorporating style transfer to default SSL augmentation pipelines.

**Our contributions:**

- We propose *SASSL*, a novel data augmentation technique based on Neural Style Transfer that naturally preserves semantic information while varying stylistic information (Section 4).

- We empirically demonstrate improved downstream task performance by incorporating our method into MoCo v2, without additional hyperparameter tuning. We report more than 2% top-1 accuracy improvement on the challenging ImageNet dataset (Section 5.1).

- We show our augmentation method learns more robust representations by measuring their transfer learning capabilities on five diverse datasets. SASSL improves linear probing performance by up to 3.75% and fine-tuning by up to 1% on out-of-distribution tasks (Section 5.2).

- We observe that balancing stylization using representation blending and image interpolation performs best, and adding external style datasets for transfer can improve performance (Section 5.3).

## 2 RELATED WORK

### 2.1 DATA AUGMENTATION IN SSL

Typical data augmentation methods in SSL applied to vision tasks correspond to image cropping and resizing, flipping, rotation, color augmentation, noise addition, and solarization. Examples of methods using these are MoCo (He et al., 2020), SimCLR (Chen et al., 2020a), BYOL (Grill et al., 2020), and SimSiam (Chen & He, 2021), among others. Other research has explored how to select augmentation hyperparameters to learn more robust and general features (Wagner et al., 2022; Reed et al., 2021; Tian et al., 2020). In contrast, our method proposes a new preprocessing technique that can be incorporated in any existing augmentation pipeline, obtaining improved performance without additional hyperparameter tuning.

Novel augmentation techniques have been proposed for both supervised and self-supervised learning. For example, Han et al. (2022) propose a novel technique to apply multiple data augmentation techniques in a local fashion by splitting the image into patches. This is aligned with other local augmentation techniques such as random erasing (Zhong et al., 2020), CutMix (Yun et al., 2019), Mixup (Zhang et al., 2017), among others. While these methods focus on masking or severely

distorting image patches, disregarding their semantic information, our approach is a global transformation that decouples an image into its appearance and semantic attributes. This allows applying transformations exclusively to its appearance without affecting the image semantics.

## 2.2 NEURAL STYLE TRANSFER

Recent image generation and style transfer algorithms (Liu et al., 2021; Heitz et al., 2021; Jing et al., 2020; Yoo et al., 2019; Risser et al., 2017; Gatys et al., 2017; Chen et al., 2021a; Wang et al., 2020) use deep learning, such as CNNs, to measure the texture similarity between two images in terms of the distance between their neural activations. Specifically, feature maps of a pre-trained classifier such as VGG-19 (Simonyan & Zisserman, 2014) are extracted and low-order moments of their distribution (Kessy et al., 2018; Huang & Belongie, 2017; Sheng et al., 2018) are used as texture descriptors. By matching such feature statistics, CNN-based techniques have shown promising results transferring texture between arbitrary images, significantly improving over classic texture synthesis approaches (Portilla & Simoncelli, 2000; Zhu et al., 2000; Heeger & Bergen, 1995).

A large body of work in CNN-based style transfer focuses on *artistic* applications such as reproducing an artwork style and imposing it over a scene of interest. These methods adopt either (i) an iterative optimization approach (Heitz et al., 2021; Risser et al., 2017; Gatys et al., 2016; 2017; Li et al., 2017a), where an initial guess is gradually transformed to depict a style of interest or (ii) an encoding-decoding strategy (Liu et al., 2021; Jing et al., 2020; Li et al., 2017b; Chen et al., 2021a; Wang et al., 2020), where one or more CNN-based image generators are trained to impose a target texture over the scene of interest in a single forward pass. While selecting one of these implies a trade-off between synthesis quality and computational cost, in most cases the generated stylization shows an unnatural appearance, *i.e.*, it does not capture most of the texture details of a style image.

Alternative techniques show impressive style generalization with much simpler transformations. In their seminal work, Dumoulin et al. (2017) propose a style transfer technique that extracts a low-dimensional representation from an arbitrary style image. This representation is then used to predict scaling and shifting values that are used to normalize the feature maps of a content image. The authors empirically show that, by learning such a contracted representation of style images, their method can represent styles from a wide range of unseen domains, leading to a faithful stylization.

Other works have applied style transfer in context of supervised learning, to produce additional labelled data (Zheng et al., 2019), remove model biases (Geirhos et al., 2018) and improve robustness against adversarial attacks (Hong et al., 2021). To our knowledge, this is the first work using neural style transfer to learn more performant self-supervised representations, without labelled data.

## 3 PRELIMINARIES

### 3.1 SELF-SUPERVISED LEARNING

Our proposed work deals with data augmentation in SSL. More precisely, we propose a novel pre-processing operation to enhance representation learning. Before introducing our work, we review the fundamentals of SSL by revisiting a well established technique: SimCLR.

**A Simple Framework for Contrastive Learning.** *SimCLR* learns compressed representations by maximizing the agreement between differently augmented views of the same data example in a latent space, while minimizing the similarity to different examples. Potential augmentations include random cropping, flipping, color jitter, blurring, and solarization. By maximizing the similarity of augmented training samples, the network learns to create robust representations that are invariant to simple distortions that could occur in the real world, and which should not affect the semantic content of an image. By simultaneously contrasting against other examples, the representations learn to extract information that is unique to each example. The combination of the two leads to the extraction of useful semantic information from the data.

Given a batch of $N$ input images $\{i_k\}_{k=1}^{N}$, $2N$ augmented samples are generated by applying distinct transformations to each image. These transformations correspond to the same data augmentation pipeline. Let $R$ correspond to all possible augmentations. Then, positive pairs correspond to augmented views of the same input sample, and negative pairs correspond to views coming from

different input images. Based on this, the $2N$ augmented samples $\{\tilde{\boldsymbol{i}}_l\}_{l=1}^{2N}$ can be organized so that indices $l = 2k - 1$ and $l = 2k$ correspond to views of the same input sample

$$\tilde{\boldsymbol{i}}_{2k-1} = r(\boldsymbol{i}_k), \quad \tilde{\boldsymbol{i}}_{2k} = \hat{r}(\boldsymbol{i}_k), \quad \hat{r}, r \sim R \tag{1}$$

Once augmented views are obtained, a representation is computed using an encoder (typically a CNN-based model). The representations are then fed to a projection head which further compresses the representation into a lower-dimensional manifold where different views of the same image are close together and those from different images are far apart. Let $h$ and $g$ be the encoder (e.g. a ResNet-50 backbone) and projection head (e.g. a multilayer perceptron), respectively. Then, embeddings are obtained for each augmented sample

$$\boldsymbol{z}_l = g \circ h(\tilde{\boldsymbol{i}}_l) \tag{2}$$

SimCLR relies on the normalized temperature-scaled cross entropy loss (NT-Xent) to learn how to distinguish between positive pairs of augmented samples. First, the cosine similarity distance between every pair of embeddings is computed

$$s_{m,n} = \frac{\langle \boldsymbol{z}_m, \boldsymbol{z}_n \rangle}{\|\boldsymbol{z}_m\| \|\boldsymbol{z}_n\|} \tag{3}$$

The final step is to use a contrastive loss to train the model. The contrastive loss compares the embeddings of positives and encourages them to be as similar as possible in the latent space. Since the loss is normalized, it naturally forces the representations of views from two different images (negatives) to be far away from each other.

$$\mathcal{L} = \frac{1}{2N} \sum_{k=1}^{N} \left[ l(2k-1, 2k) + l(2k, 2k-1) \right] \tag{4}$$

$$l(m, n) = -\log \left( \frac{\exp(s_{m,n}/\tau)}{\sum_{l=1}^{2N} \mathbb{1}_{m \neq n} \exp(s_{m,l}/\tau)} \right) \tag{5}$$

where $\tau \in \mathbb{R}_{++}$ is the temperature factor and $\mathbb{1}$ the indicator function. While SimCLR is a simple framework, it pushed the state-of-the-art significantly on a wide range of downstream tasks including image classification, object detection, and semantic segmentation. Follow up works to SimCLR such as Grill et al. (2020); Chen & He (2021); Caron et al. (2020; 2021); Assran et al. (2022); Zbontar et al. (2021); Chen et al. (2020b; 2021b); Bardes et al. (2021) have proposed modifications to this setup which attempt to further improve the downstream task performance.

## 3.2 NEURAL STYLE TRANSFER

Neural Style Transfer algorithms aim to preserve the content of an image while imposing the style of another. These approaches assume that the statistics of shallower layers of a trained CNN encode style, while deeper layers encode content. Thus, early Style transfer algorithms worked by passing the pair of *content* and *style* images to a trained image encoder and optimizing the content image to produce activations which had similar statistics to the style image at shallower layers while keeping the deeper layer activations unchanged (Simonyan & Zisserman, 2014).

Once semantic and texture features are extracted, a *stylized* image with the semantic attributes of the content image and the texture properties of the style image is generated. The two main approaches are *image-optimization based* and *model-based* methods. In what follows, we introduce a model-based Style Transfer technique adopted by our proposed method due on its generalization and efficiency properties. Jing et al. (2019) provide an in-depth study of image stylization techniques.

**Fast Style Transfer.** Dumoulin et al. (2017) proposed an arbitrary style transfer method with remarkable generalization properties. Their algorithm, *Fast Style Transfer*, has been shown to accurately represent unseen artistic styles by training a model to predict first and second moments of latent image representations at multiple scales. Such moments are used as arguments of a special form of instance normalization, denominated *conditional instance normalization* (CIN), to impose style over arbitrary input images. Below we formally introduce the Fast Style Transfer algorithm.

Given a content image $i_c \in \mathbb{R}^{C \times H_c \times W_c}$ and a style image $i_s \in \mathbb{R}^{C \times H_s \times W_s}$, the produced stylized image $i_{cs}$ corresponds to

$$i_{cs} = \mathcal{T}(i_c, z_s) \in \mathbb{R}^{C \times H_c \times W_c} \qquad (6)$$

where $\mathcal{T}$ is a stylization network and $z_s$ is a feature embedding extracted from the style image using a pre-trained feature extractor denoted as $\mathcal{F}$, e.g., InceptionV3 (Szegedy et al., 2016).

$$z_s = \mathcal{F}(i_s) \in \mathbb{R}^D \qquad (7)$$

Here, we assume $z_s$ to be a contracted embedding of the style image ($D \ll H_s W_s$). The stylization network $\mathcal{T}$ is comprised by $L$ blocks $\{t_l\}_{l=1}^L$. These extract high-level features from the content image, align them to the style embedding $z_s$ and map the resulting features back to the pixel domain.

The style of $i_s$ encapsulated in $z_s$ is transferred to the content image using CIN. This is applied to a particular set of layers to impose the target texture and color scheme by aligning feature maps at different scales. We define the set of layers where CIN is applied as $\mathcal{A}$. conditional instance normalization (CIN) consists of an extended form of instance normalization where the target mean and standard deviation are extracted from an arbitrary style representation $z$. Given an input $i \in \mathbb{R}^{C \times H \times W}$ and a style representation $z \in \mathbb{R}^D$, CIN is defined as

$$\hat{i} = \mathrm{CIN}(i, z)$$

$$\hat{i}^{(k)} = \gamma^{(k)}(z)\left(\frac{i^{(k)} - \mathbb{E}[i^{(k)}]}{\sigma(i^{(k)})}\right) + \lambda^{(k)}(z), \; k \in \{1, \ldots, C\}$$

where $i^{(k)}$ corresponds to the $k$-th input channel, and the sample mean and standard deviation are computed along its spatial support. Here, $\gamma^{(k)}, \lambda^{(k)} : \mathbb{R}^D \mapsto \mathbb{R}$ are trainable functions that predict scaling and offset values from the latent representation $z$ for the $k$-th input channel. Following this, the layers in the stylization network are characterized by

$$\phi_{c,l} = \begin{cases} \mathrm{CIN}_l\big(t_l(\phi_{c,l-1}), z_s\big), & l \in \mathcal{A} \\ t_l(\phi_{c,l-1}), & l \notin \mathcal{A} \end{cases} \qquad (8)$$

where the network input corresponds to $\phi_{c,0} = i_c$. We use the subscript $l$ in the CIN operation to indicate that each layer has its own $\gamma$ and $\lambda$ functions to normalize feature maps independently.

Our method uses the Fast Style Transfer algorithm. Given its generalization properties and low-dimensional style representations, it is a good match for our framework, where style representations from multiple domains must be extracted, manipulated and transferred in an efficient manner.

## 4 Proposed Method: SASSL

Recent work in self-supervised learning has shown the importance of diverse data augmentation to achieve state-of-the-art performance across downstream tasks. However, few approaches focus on transformations that preserve the semantic information of pre-training data, though this is essential for strong self-supervised representations. To tackle this limitation, we propose *SASSL*, a novel data-augmentation strategy based on neural style transfer that decouples semantic from appearance attributes, enforcing transformations that operate strictly on the input color and texture.

**Style Transfer as data preprocessing.** We incorporate style transfer to the default preprocessing pipeline of SSL methods. We note that our method is not specific to a particular SSL method, and can be readily applied with different methods. Figure 1 describes an example of our SASSL data augmentation pipeline, where Style Transfer is applied after random cropping. A raw input image $i_0$ is randomly cropped, producing a view that is taken as the content image $i_c$. Given an arbitrary style image $i_s$ (we discuss the choice of $i_s$ below), the style transfer block generates a stylized image $i_{cs}$ by imposing the texture attributes of $i_s$ over $i_c$. Finally, the stylized image $i_{cs}$ is passed to the remaining data augmentation blocks to produce an augmented sample $i_{aug}$.

As discussed in recent work on self-supervised augmentation (Han et al., 2022; Chen et al., 2020a), adding a strong transformation to a self-supervised method tends to degrade performance. For this reason, it is crucial to control the amount of stylization imposed in the augmentation stage. We

---

**Algorithm 1:** Style transfer augmentation block

---

**Input:** $\boldsymbol{i}_c, \boldsymbol{i}_s, \mathcal{F}, \mathcal{T}, \alpha_{\min}, \alpha_{\max}, \beta_{\min}, \beta_{\max}$
**Output:** $\boldsymbol{i}_{\text{cs}}$

---

$\boldsymbol{z}_c \leftarrow \mathcal{F}(\boldsymbol{i}_c)$ ;                  # Style representation of content image
$\boldsymbol{z}_s \leftarrow \mathcal{F}(\boldsymbol{i}_s)$ ;                  # Style representation of style image

$\alpha \sim \mathcal{U}(\alpha_{\min}, \alpha_{\max})$;                  # Blending factor
$\hat{\boldsymbol{z}} \leftarrow (1 - \alpha)\boldsymbol{z}_c + \alpha\boldsymbol{z}_s$;                  # Feature blending
$\hat{\boldsymbol{i}}_{\text{cs}} \leftarrow \mathcal{T}(\boldsymbol{i}_c, \hat{\boldsymbol{z}})$

$\beta \sim \mathcal{U}(\beta_{\min}, \beta_{\max})$;                  # Interpolation factor
$\boldsymbol{i}_{\text{cs}} \leftarrow (1 - \beta)\boldsymbol{i}_c + \beta\hat{\boldsymbol{i}}_{\text{cs}}$;                  # Stylized image

---

do so by introducing three hyperparameters: a stylization probability $p \in [0, 1]$, which dictates whether an image is stylized or not, a blending factor $\alpha \in [0, 1]$ to combine content $\boldsymbol{z}_c$ and style $\boldsymbol{z}_s$ representations, and an interpolation factor $\beta \in [0, 1]$ to combine content $\boldsymbol{i}_c$ and stylized $\boldsymbol{i}_{\text{cs}}$ images.

Given style representations extracted from content and style reference images $\boldsymbol{z}_c = \mathcal{F}(\boldsymbol{i}_c)$ and $\boldsymbol{z}_s = \mathcal{F}(\boldsymbol{i}_s)$, respectively, we obtain an intermediate stylized image $\hat{\boldsymbol{i}}_{\text{cs}}$ by applying a convex combination on them based on the blending factor $\alpha$.

$$\hat{\boldsymbol{i}}_{\text{cs}} = \mathcal{T}(\boldsymbol{i}_c, \hat{\boldsymbol{z}}) \tag{9}$$

$$\hat{\boldsymbol{z}} = (1 - \alpha)\boldsymbol{z}_c + \alpha\boldsymbol{z}_s \tag{10}$$

Then, the final stylization output is obtained via a convex combination between the intermediate stylized image $\hat{\boldsymbol{i}}_{cs}$ and the content image $\boldsymbol{i}_c$ based on the interpolation factor $\beta$.

$$\boldsymbol{i}_{\text{cs}} = (1 - \beta)\boldsymbol{i}_c + \beta\hat{\boldsymbol{i}}_{\text{cs}} \tag{11}$$

Algorithm 1 describes our proposed style transfer data augmentation block. Figure 2 illustrates the effect of the feature blending and image interpolation operations, showcasing their importance to control the stylization effect without degrading the semantic attributes.

SASSL operates over minibatches, allowing efficient data preprocessing. Let $\boldsymbol{I}_c \in \mathbb{R}^{B \times C \times H_c \times W_c}$ and $\boldsymbol{I}_s \in \mathbb{R}^{B \times C \times H_s \times W_s}$ be content and style minibatches, respectively, comprised by $B$ images $\boldsymbol{I}_c^{(b)}$ and $\boldsymbol{I}_s^{(b)}, b \in \{1, \ldots, B\}$. Then, the stylized minibatch $\boldsymbol{I}_{\text{cs}} \in \mathbb{R}^{B \times C \times H_c \times W_c}$ is generated by applying style transfer between a sample from the content batch and a sample from a style batch, given an arbitrary selection criterion. In what follows, we propose two alternatives for selecting style images to balance between augmentation diversity and efficiency.

**Diversifying style references.** In contrast to traditional data augmentation, style transfer can leverage a second dataset to extract style references. This opens the possibility of selecting style images from different domains, diversifying the transformations applied to the pre-training dataset. *SASSL* relies on two approaches for sampling style references: *external* and *in-batch* stylization.

*External* stylization consists on pre-computing representations of an arbitrary style dataset and sampling from them during pre-training. This allows controlling the styles to impose on the augmented views while reducing the computational overhead of Style Transfer. Under this configuration, the Style Transfer block receives a content minibatch along with a minibatch of pre-computed style representations extracted from an arbitrary style dataset, and generates a minibatch of stylized images.

On the other hand, *in-batch* stylization uses the styles depicted in the content dataset itself by using other images of the content minibatch as style references. This is of particular interest for large-scale pre-training datasets covering multiple image categories and thus textures (e.g. ImageNet). So, enabling the use of a single dataset for both pre-training and stylization is a valid alternative.

Let $\boldsymbol{I}_c \in \mathbb{R}^{B \times C \times H_c \times W_c}$ be a pre-training minibatch comprised by $B$ images and $\boldsymbol{I}_c^{(b)} \in \mathbb{R}^{C \times H_c \times W_c}, b \in \{1, \ldots, B\}$ the image in the $b$-th index. Then, samples from the same minibatch can be used as style references by associating pairs of images in a circular fashion. More

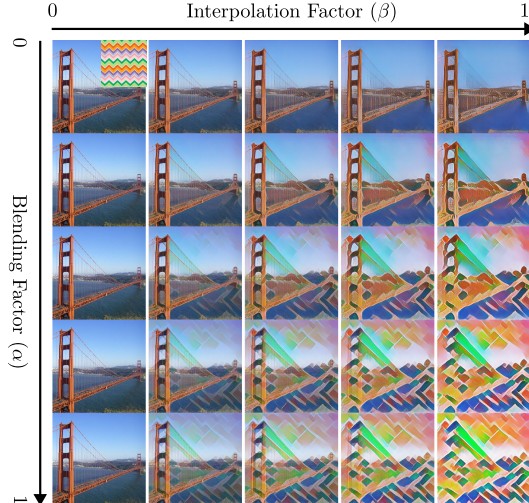

Figure 2: **Feature blending and image interpolation**. A fine-grained control over the final stylization is obtained by introducing interpolation factors $\alpha$ and $\beta$ to operate in the feature space and pixel domain, respectively.

Figure 3: **Style transfer examples**. Stylized images generated using style references from the same domain (**in-batch**) as well as from other domain (**external**). Stylization obtained using a blending factor $\alpha = 0.5$.

precisely, a style minibatch $\boldsymbol{I}_s \in \mathbb{R}^{B \times C \times H_c \times W_c}$ is generated by applying a circular shift on the content minibatch indices

$$\boldsymbol{I}_s^{(b)} = \boldsymbol{I}_c^{((b-b_0) \bmod B)} \tag{12}$$

where mod denotes the modulo operation and $b_0$ is an arbitrary offset. Figure 3 shows ImageNet samples stylized using pre-computed features from Painter by Numbers via *external* stylization, as well as using other ImageNet samples taken from the content minibatch via *in-batch* stylization.

## 5 EXPERIMENTS

### 5.1 DOWNSTREAM TASK PERFORMANCE

We compare the downstream ImageNet classification accuracy of SSL models learned using SASSL against a MoCo v2 (Chen et al., 2020b) baseline learned using the default data augmentation. Note that MoCo v2 and SimCLR are extremely similar (same loss, backbone architecture, and default augmentations), and differ mainly by MoCo's use of a momentum encoder.

**Pre-training settings.** Our pre-training setup is similar to the canonical SSL setup used to pre-train SimCLR and BYOL. We use the same loss, architecture, optimizer, and learning rate schedule as MoCo v2 for fair comparison. We pre-train a ResNet-50 encoder on ImageNet for $1,000$ epochs using our proposed method. Once the backbone is pre-trained, to measure downstream accuracy, we add a linear classification head on top of the backbone and train in a supervised fashion on ImageNet.

SASSL applies Style Transfer only to the left view during pre-training, *i.e.*, no changes in augmentation are applied to the right view. Style transfer is applied with a probability $p = 0.8$ using blending and interpolation factors drawn from a uniform distribution $\alpha, \beta \sim \mathcal{U}(0.1, 0.3)$. We found that this modest application of style transfer best complimented the existing augmentations, avoiding overly-strong augmentations that can reduce performance (Han et al., 2022; Chen et al., 2020a).

**Results.** Table 1 compares the downstream classification accuracy obtained by our SASSL augmentation approach on MoCo v2 using *external* stylization from the Painter by Numbers dataset (Kan, 2016). Results indicate our proposed augmentation improves downstream task performance by 2.09% top-1 accuracy. This highlights the value of Style Transfer augmentation in self-supervised training, where downstream task performance significantly boosts by incorporating transformations that decouple content and style. We also report results with *in-batch* stylization in Section 5.3.

Table 1: **Downstream classification performance on ImageNet**. Classification accuracy of MoCo v2 pre-trained using our proposed style transfer augmentation approach (SASSL). Mean and standard deviation reported over five randomly initialized models.

| Method | Top-1 Acc. (%) | Top-5 Acc. (%) |
|---|---|---|
| MoCo v2 (default) | $72.55 \pm 0.67$ | $91.19 \pm 0.34$ |
| SASSL + MoCo v2 (**Ours**) | $\mathbf{74.64 \pm 0.43}$ | $\mathbf{91.68 \pm 0.36}$ |

## 5.2 TRANSFER LEARNING PERFORMANCE

To understand robustness and generalization of representations learned using our approach, we study transfer learning performance across five tasks. By incorporating Style Transfer, we hypothesize that learned representations become invariant to changes in appearance (e.g. color and texture). This forces the feature extraction to rely exclusively on semantic attributes. As a result, the learned representations may become more robust to domain shifts, improving downstream task performance across datasets. We empirically show this by evaluating the transfer learning performance of representations trained using SASSL.

**Downstream training settings.** We compare the transfer learning performance of ResNet-50 pre-trained via MoCo v2 using SASSL against a vanilla MoCo v2 approach using default augmentations. Models are pre-trained on ImageNet and transferred to four target datasets: ImageNet 1% subset (Chen et al., 2020a), iNaturalist '21 (iNat21) (iNaturalist 2021), Diabetic Retinopathy Detection (Retinopathy) (Kaggle & EyePacs, 2015), and Describable Textures Dataset (DTD) (Cimpoi et al., 2014). We also evaluate downstream performance on ImageNet, giving a total of five target datasets.

To have a clear idea of the effect of the style dataset in the SASSL pipeline, we pre-train five ResNet-50 backbones, each using a different style source. We cover the following style datasets: ImageNet, iNat21, Retinopathy, DTD, and Painter by Numbers (PBN). More precisely, we transfer five models, each pre-trained on a different style, to each of five target datasets. We also transfer a model pre-trained using the default augmentation pipeline. This leads to 30 different transfer learning scenarios used to better understand the effect of various styles on different image domains.

Transfer learning performance is evaluated in terms of top-1 classification accuracy on linear probing and fine-tuning. All models were pre-trained as described in Section 5.1. We report mean and standard deviation across five model initializations. Please refer to Section A.4 for linear probing and fine-tuning training and testing settings.

**Results.** Table 2 shows the top-1 classification accuracy obtained via transfer learning. For linear probing, SASSL significantly improves the average performance on four out of five target datasets by up to 3.75% top-1 classification accuracy. For Retinopathy, our augmentation method obtains on-par linear probing performance with respect to the default MoCo v2 model.

Similarly, for fine-tuning, all models trained via SASSL outperform the baseline. Results show the average top-1 classification accuracy improves by up to 1.32%. This suggests SASSL generalizes across datasets, spanning from texture (DTD) to the medical images (Retinopathy). Note that, for a fair comparison, we do not perform hyperparameter tuning on the default augmentations.

## 5.3 ABLATION STUDY

To better understand the impact of each Style Transfer component in our augmentation pipeline, we conduct an ablation study over the hyperparameters of our style transfer model. Similar to previous experiments, we quantify our uncertainty by reporting the mean and variance obtained by five independent trials.

**Downstream training settings.** For the ablation study, we cover four scenarios: (i) default MoCo v2, (ii) SASSL + MoCo v2 using only in-batch representation blending, *i.e.*, without interpolation ($\beta = 1$), (iii) SASSL + MoCo v2 using in-batch representation blending and pixel interpolation, and (iv) SASSL + MoCo v2 using all its attributes (representation blending, pixel interpolation and an external style dataset).

**Results.** Table 3 shows our ablation study on MoCo v2. It highlights the importance of controlling the amount of stylization using both representation blending and image interpolation. Without

Table 2: **Transfer learning performance.** Downstream top-1 classification accuracy of MoCo v2 pre-trained on ImageNet using SASSL. SASSL generates specialized representations that improve transfer learning performance, both in linear probing and fine-tuning. Mean and standard deviation reported over five randomly initialized models.

| | | Target Dataset | | | |
|---|---|---|---|---|---|
| | | ImageNet | ImageNet (1%) | iNat21 | Retinopathy | DTD |
| | | *Linear Probing* | | | | |
| Style Dataset | None (Baseline) | $72.55 \pm 0.67$ | $53.23 \pm 0.45$ | $41.33 \pm 0.06$ | $\mathbf{75.92 \pm 0.12}$ | $72.68 \pm 0.7$ |
| | ImageNet (**Ours**) | $74.07 \pm 0.46$ | $56.87 \pm 0.43$ | $45.01 \pm 0.37$ | $75.73 \pm 0.12$ | $73.69 \pm 1.22$ |
| | iNat21 (**Ours**) | $74.28 \pm 0.38$ | $56.76 \pm 0.23$ | $44.70 \pm 0.37$ | $75.75 \pm 0.17$ | $72.75 \pm 1.01$ |
| | Retinopathy (**Ours**) | $74.07 \pm 0.62$ | $\mathbf{56.99 \pm 0.26}$ | $44.9 \pm 0.16$ | $75.78 \pm 0.08$ | $73.73 \pm 0.57$ |
| | DTD (**Ours**) | $74.32 \pm 0.32$ | $56.77 \pm 0.36$ | $\mathbf{45.08 \pm 0.31}$ | $75.76 \pm 0.11$ | $\mathbf{74.41 \pm 1.39}$ |
| | PBN (**Ours**) | $\mathbf{74.64 \pm 0.43}$ | $56.76 \pm 0.23$ | $44.70 \pm 0.37$ | $75.75 \pm 0.17$ | $72.75 \pm 1.01$ |
| | | *Fine-tuning* | | | | |
| | None (Baseline) | $74.89 \pm 0.67$ | $51.61 \pm 0.13$ | $77.92 \pm 0.14$ | $78.87 \pm 0.2$ | $71.54 \pm 0.43$ |
| | ImageNet (**Ours**) | $75.52 \pm 0.28$ | $51.74 \pm 0.14$ | $79.21 \pm 0.07$ | $79.64 \pm 0.16$ | $\mathbf{72.31 \pm 1.87}$ |
| | iNat21 (**Ours**) | $\mathbf{75.58 \pm 0.47}$ | $\mathbf{51.86 \pm 0.3}$ | $79.19 \pm 0.12$ | $79.6 \pm 0.23$ | $71.35 \pm 1.58$ |
| | Retinopathy (**Ours**) | $75.52 \pm 0.64$ | $51.76 \pm 0.26$ | $79.23 \pm 0.05$ | $79.63 \pm 0.13$ | $72.07 \pm 1.61$ |
| | DTD (**Ours**) | $75.24 \pm 0.65$ | $51.73 \pm 0.14$ | $\mathbf{79.24 \pm 0.08}$ | $\mathbf{79.7 \pm 0.15}$ | $70.59 \pm 1.42$ |
| | PBN (**Ours**) | $75.05 \pm 0.69$ | $51.85 \pm 0.16$ | $79.2 \pm 0.1$ | $79.63 \pm 0.13$ | $71.39 \pm 0.96$ |

Table 3: **Ablation Study.** Linear probing accuracy for representations trained under different configurations. Mean and standard deviation reported over five randomly initialized models.

| Augmentation | Configuration | Style Dataset | Top-1 Acc. (%) | Top-5 Acc. (%) |
|---|---|---|---|---|
| Baseline | − | − | $72.55 \pm .67$ | $91.19 \pm 0.34$ |
| SASSL (**Ours**) | Probability: $p = 0.8$, Blending: $\alpha \in [0.1, 0.3]$ Interpolation $\beta = 1$ | ImageNet (*in-batch*) | $70.87 \pm 0.72$ | $89.33 \pm 0.34$ |
| | Probability: $p = 0.8$, Blending: $\alpha \in [0.1, 0.3]$ Interpolation: $\beta \in [0.1, 0.3]$ | ImageNet (*in-batch*) | $74.07 \pm 0.62$ | $91.58 \pm 0.18$ |
| | Probability: $p = 0.8$, Blending: $\alpha \in [0.1, 0.3]$ Interpolation: $\beta \in [0.1, 0.3]$ | PBN (*external*) | $\mathbf{74.64 \pm 0.43}$ | $\mathbf{91.68 \pm 0.36}$ |

image interpolation, using style transfer as augmentation degrades the downstream classification performance by more than $1.5\%$ top-1 accuracy.

On the other hand, by balancing the amount of stylization via blending and interpolation, SASSL boosts performance by more than $1.5\%$. This is a significant improvement for the challenging ImageNet scenario. Finally, by incorporating an external dataset such as Painters by Numbers, we further improve downstream task performance by almost $2.1\%$ top-1 accuracy. This shows the importance of diverse style references and their effect on downstream tasks.

# 6 CONCLUSION

We propose SASSL, a novel data augmentation approach based on Neural Style Transfer that exclusively transforms the style of training samples, diversifying data augmentation during pre-training while preserving semantic attributes. We empirically show our approach outperforms well-established methods such as MoCo v2 by more than $2\%$ top-1 classification accuracy on the challenging ImageNet dataset. Importantly, SASSL also improves the transfer capabilities of learned representations, enhancing linear probing and fine-tuning classification performance across image domains by more than $3\%$ and $1\%$ top-1 accuracy, respectively. Future work includes extending our technique to different SSL methods and encoder architectures (e.g. ViTs), which can readily be combined with our method.

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
