# OpenReview forum: "SASSL: Enhancing Self-Supervised Learning via Neural Style Transfer"
_ICLR.cc/2024/Conference — Submitted to ICLR 2024_

### Official Review · Reviewer_4joa · 2023-10-23

**Soundness:** 2 fair
**Presentation:** 3 good
**Contribution:** 3 good
**Rating:** 6
**Confidence:** 3

**Summary:**

In this work, the authors propose an augmentation technique for SSL methods, capable of extracting styles from in-batch or external datasets and transferring them to other images. This process effectively diversifies the styles of the original images. Additionally, the authors emphasize the necessity of preserving the semantic information of pre-training data. They introduce an intermediate stylized image to prevent substantial alteration of its semantic content. The authors report significant improvements compared to baselines, underscoring the efficacy of the proposed augmentation.

**Strengths:**

1. The introduction of style augmentation is novel and holds promise for integration with other SSL methods.

2. The proposed method markedly enhances the performance of the baseline.

3. The paper is well-structured and effectively presented.

**Weaknesses:**

1. While the 2% improvement for MoCo V2 is noteworthy, the generalizability of the proposed augmentation technique necessitates evaluation across various methods, such as BYOL and SimSiam.

2. To evaluate the representation quality, more downstream experiments are needed. The authors should consider expanding their experiments such as object detection and segmentation.

3. The concept of controlling augmentation to preserve semantic information is not new. It's crucial for the authors to position their work within the context of existing literature.

[1] Demystifying Contrastive Self-Supervised Learning: Invariances, Augmentations and Dataset Biase. NeurIPS 2020.
[2] Improving Transferability of Representations via Augmentation-Aware Self-Supervision. NeurIPS 2021.
[3] RSA: Reducing Semantic Shift from Aggressive Augmentations for Self-supervised Learning. NeurIPS 2022.

**Questions:**

1. The enhancements observed during fine-tuning are not as pronounced as those in linear probing. Could the authors elucidate the factors contributing to this disparity?

2. Have the authors employed strong augmentations during the linear probing phase? Typically, in settings of SimCLR and MoCo V2, only week augmentations are utilized. It would be help present the linear probing results on ImageNet-1K without strong augmentations.

---

> ### Author Response · Authors · 2023-11-21
> **Response to Reviewer 4joa**
>
> We thank the reviewer for the valuable comments and feedback.
>
> >### 1. SASSL performance on other SSL techniques.
>
> As suggested, we conducted additional experiments to assess SASSL's performance across alternative SSL methods, namely SimCLR and BYOL. Our results consistently show performance improvements across all methods. For details on these experiments, please refer to our General Response (item 2).
>
> >### 2. Additional downstream tasks
>
> We value the reviewer's suggestion to assess our method on additional downstream tasks. We conducted further experiments on few-shot learning. The results from one-shot and ten-shot experiments on ImageNet reveal that the representations acquired using SASSL improve performance in both scenarios. For the details, please refer to our General Response (item 3).
>
> >### 3. Comparison to work on data augmentation
>
> In what follows, we compare our proposed method and the work suggested by the reviewer. Please refer to Appendix Sec. A.9 for a more in-depth comparison.
>
> Purushwalkam et al [1] suggest an augmentation strategy based on natural (temporal) transformations that occur in videos as an alternative to learning from occluded (cropped) images from object-centric datasets. In contrast, our approach utilizes neural style transfer to decouple images into content and style, allowing us to apply transformations exclusively to the style of pre-training samples while preserving the content information. Their approach relies on datasets generated on video frames to incorporate temporal invariance, while SASSL can be used on standard image datasets, enabling its integration to default augmentation pipelines.
>
> Lee et al. [2] propose to include an auxiliary loss for SSL methods to capture the difference between augmented views, leading to better downstream performance on datasets where the task relies on information that may have been lost due to invariances introduced by aggressive augmentation. While their method modifies the pre-training loss and requires the use of data augmentation parameters to learn representations, SASSL does not require any additional loss components or auxiliary information from each applied augmentation to function. Our method complements the default data augmentation pipeline with a content-preserving transformation to obtain more general representations.
>
> Bai et al. [3] propose an alternative SSL augmentation pipeline to prevent the loss of semantic information in the learned representations by encouraging learning from weakly augmented views and gradually transitioning to strongly augmented views. The method avoids semantic shifts caused by aggressive image transformations by leveraging the tendency of SSL models to memorize clean views during early pre-training stages and by proposing a multi-stage augmentation pipeline to generate weak and strongly augmented views. Differently, our approach does not require additional augmented views or dynamic control over the loss weights during training. SASSL utilizes a content-preserving transformation that only modifies stylistic characteristics (color and texture), leaving the natural image structure and objects unchanged. Our approach incorporates a transformation into the default augmentation pipeline without altering the loss function or instance discrimination strategy.
>
> >### 4. Linear Probing vs. Fine-Tuning improvement gap
>
> The difference in performance between linear probing and fine-tuning arises from how the representation model is treated. Linear probing involves training a classification head atop representations extracted through SASSL. In contrast, fine-tuning entails fully training both the representation and classification models on the target dataset. Consequently, fine-tuning diminishes the impact of style transfer employed during pre-training by retraining the backbone, lessening its effect on downstream performance.
>
> Despite this performance gap, SASSL demonstrates improved performance in both linear probing and fine-tuning. This can be attributed to (i) more robust image representations (evidenced by the linear probing results) and (ii) better model initialization (supported by the fine-tuning results). It's noteworthy that alternative SSL methods also show less improvement over the baseline during fine-tuning. For instance, BYOL exhibits this behavior compared to SimCLR, as shown in Table 3 of BYOL's paper [r3].
>
> [r3] Grill, Jean-Bastien, et al. "Bootstrap your own latent-a new approach to self-supervised learning." Advances in neural information processing systems 33 (2020): 21271-21284.
>
> >### 5. Data augmentation during linear probing
>
> SASSL employs the same data augmentation and optimizer settings for both linear probing and fine-tuning as MoCo v2, which correspond to **weak data augmentations**. For a more detailed description of these settings, please refer to Appendix Sec. A.4.
>
> Please note that SASSL utilizes style transfer only during pre-training, not during downstream training.

---

### Official Review · Reviewer_ifUN · 2023-10-31

**Soundness:** 3 good
**Presentation:** 3 good
**Contribution:** 2 fair
**Rating:** 6
**Confidence:** 3

**Summary:**

This paper introduces a novel data augmentation technique called SASSL (Style Augmentations for Self Supervised Learning) based on Neural Style Transfer. SASSL decouples semantic and stylistic attributes in images and applies transformations exclusively to the style while preserving content, generating diverse augmented samples that better retain their semantic properties. Experimental results demonstrate that SASSL improves classification performance on ImageNet and transfer learning performance on diverse datasets compared to existing methods. The paper also provides an overview of self-supervised learning, contrastive learning, style transfer, and texture synthesis, discussing the challenges and limitations of existing methods in these areas. The contributions of this paper include the introduction of the SASSL data augmentation technique, which effectively combines neural style transfer with self-supervised learning, and the demonstration of its benefits in improving classification and transfer learning performance.Their proposed augmentation improves downstream task performance by 2.09% top-1 accuracy on imagenet dataset and improves linear probing performance by up to 3.75% and fine-tuning by up to 1% on transfer learning on five diverse datasets. The paper also highlights the potential for future extensions of SASSL to other self-supervised learning methods and encoder architectures.

**Strengths:**

In terms of originality, the paper introduces a novel data augmentation technique called SASSL that combines existing ideas like neural style transfer with self-supervised learning to increase robustness and have better transferability. This approach of decoupling semantic and stylistic attributes in images and applying transformations exclusively to the style while preserving content is unique and innovative. It offers a fresh perspective on data augmentation in self-supervised learning and provides a new way to generate diverse augmented samples while retaining semantic properties.
The quality of the paper is evident in the thorough experimental evaluation conducted. The authors compare SASSL with existing methods and demonstrate its effectiveness in improving classification performance on ImageNet and transfer learning performance on diverse datasets. The experimental results are well-presented and provide strong evidence for the benefits of SASSL.
Clarity is another strength of the paper. They explain the motivation behind SASSL and the technical details of the approach in a clear and concise manner. The paper is well-structured and easy to follow, making it accessible to readers.
In terms of significance, the paper makes a valuable contribution to the field of self-supervised learning and data augmentation. By introducing SASSL, the authors address the challenge of preserving semantic information while applying style transformations, which is crucial for generating augmented samples that retain their original content. The improved classification and transfer learning performance achieved by SASSL highlight its potential for enhancing the generalization capabilities of learned representations. This has implications for various downstream tasks and can lead to more robust and accurate machine learning models.
Overall, the paper demonstrates originality, quality, clarity, and significance in its approach to data augmentation in self-supervised learning. It introduces a novel technique, presents the information clearly, and contributes to the advancement of the field.

**Weaknesses:**

One major concern is the lack of a detailed discussion about relevant data augmentation strategies, especially those manipulating image textures like [1]. Regarding the experiments, no existing data augmentation algorithm is compared against SASSL.

Another potential weakness of the paper is the lack of a comprehensive ablation study to analyze the effect of the number of layers in the stylization network on the performance. Conducting such an ablation study would provide deeper insights into the strengths and limitations of SASSL and help guide future improvements.
Additionally, the paper could provide more insights into the computational efficiency of the SASSL technique and comparison with other existing methods that also leverage style transfer for data augmentation. This would provide a clearer understanding of the computational requirement and unique contributions and advantages of SASSL compared to previous approaches.

[1] ImageNet-trained CNNs are biased towards texture; increasing shape bias improves accuracy and robustness. (ICLR’19).

**Questions:**

What is the need for incorporating blending and pixel interpolation ? Is it purely for experimental purposes ? What is their significance in the proposed SASSL technique?

---

> ### Author Response · Authors · 2023-11-21
> **Response to Reviewer ifUN (1/2)**
>
> We appreciate the reviewer's acknowledgment of our paper's presentation and for sharing insightful feedback.
>
> >### 1. Discussion on previous texture-based techniques
>
> We thank the reviewer for pointing out related work on image texture and its effect on perceptual tasks. While Geirhos et al. [1] studies how CNNs are biased towards texture in the context of supervised learning, we focus on preserving semantic information in self-supervised learning by exclusively distorting the style component of pre-training samples. While their main hypothesis is that CNN classifiers trained via supervised learning are biased towards texture, ours is that decoupling content and style in order to exclusively distort the style leads to stronger self-supervised image representations.
>
> While both works rely on NST, their method uses it to create a stylized dataset (Stylized-ImageNet), whereas our approach is to incorporate style transfer in the augmentation pipeline to create stylized images on-the-fly while controlling their effect via blending and interpolation hyperparameters. Experimentally, we also cover a wider set of style datasets (beyond Painter by Numbers) and downstream datasets. We emphasize that Geirhos's approach is designed for supervised learning, while SASSL focuses on representation learning using SSL. Consequently, these methods are not directly comparable (See Appendix Sec. A.9 for a more detailed comparison between techniques).
>
> >### 2. Ablation study on stylized layers
>
> At the reviewer's suggestion, we conducted an ablation study to investigate the impact of the number of layers stylized during pre-training. We examined three scenarios: (i) stylizing the first two residual blocks (four layers corresponding to Residual blocks 1 and 2), (ii) stylizing the first four residual blocks (eight layers corresponding to Residual blocks 1 to 4), and (iii) stylizing all residual layers (ten layers corresponding to Residual layers 1 to 5).
>
> For each scenario, we pre-trained and linearly probed a ResNet-50 on ImageNet using SASSL + MoCo v2 with its optimal configuration (blending and interpolation between 0.1 and 0.3, using Painter by Numbers as the external style dataset). To provide a clear comparison, we compared the classification accuracy of these three models to the model trained without feature alignment (using the stylization network as an autoencoder), and our complete SASSL + MoCo v2 model (stylizing all layers). Note that the latter stylizes all residual and upsampling blocks (a total of thirteen layers).
>
> Results indicate a gradual improvement in accuracy as the number of stylized layers increases. Stylizing only the first four layers yielded no significant improvements, resulting in approximately the same accuracy as the model pre-trained without feature alignment. Stylizing the first eight layers enhanced top-1 classification accuracy by 0.32% over the model with no feature alignment. Finally, stylizing the first ten layers improved accuracy by 0.5%. These findings suggest that deeper layers have a more pronounced impact on accuracy improvement. Additionally, the model pre-trained with full stylization achieved a 1.61% improvement over the model with no feature alignment, indicating that the stylization occurring in the upsampling layers plays a dominant role in the gains provided by SASSL.
>
> Our study suggests that significant improvements can be achieved by solely stylizing deeper layers, potentially reducing the computational demands of SASSL while still enabling the acquisition of robust image representations (See Appendix Sec. A.7 for more details).
>
> | Augmentation Settings |              Stylization Mode              | Top-1. Acc (%) | Top-5 Acc. (%) |   |
> |:---------------------:|:------------------------------------------:|:--------------:|:--------------:|---|
> |        Default        |                      -                     |      72.97     |      90.86     |   |
> |         SASSL (Ours)        | Image encoding only (no feature alignment) |      73.77     |      91.64     |   |
> |         SASSL (Ours)        |          Stylize first four layers         |      73.75     |      91.58     |   |
> |         SASSL (Ours)        |         Stylize first eight layers         |      74.09     |      91.76     |   |
> |         SASSL (Ours)        |          Stylize first ten layers          |      74.27     |      91.74     |   |
> |         **SASSL (Ours)**        |              Full stylization              |      **75.38**     |      **92.21**     |   |

---

> ### Author Response · Authors · 2023-11-21
> **Response to Reviewer ifUN (2/2)**
>
> >### 3. Computational Requirements
>
> We ran additional experiments to compare the runtime of our proposed method to the default augmentation pipeline, as suggested by the reviewer. We calculated the throughput (augmented images per second) of SASSL relative to MoCo v2's data augmentation. The throughput was determined by averaging 100 independent trials on 128 x 128-pixel images with a batch size of 2048. We also report the relative change, which is the percentage decrease in throughput compared to the default data augmentation. All experiments were carried out on a single TPU.
>
> The following table summarizes the throughput comparison. SASSL reduces throughput by about 20% due to the computational expense of stylizing large batches, which involves running a forward pass of the stylization model. However, SASSL achieves up to a 2% top-1 classification accuracy improvement, as shown empirically. We believe this to be a favorable tradeoff between performance and runtime (See Appendix Sec. A.8 for more details).
>
> | Augmentation Method | Throughput (Images/second) | Relative change (%) |   |   |
> |:-------------------:|:--------------------------:|:-------------------:|:-:|---|
> |    SimCLR (default)   |            37.45           |          -          |   |   |
> |    **SASSL + SimCLR (Ours)**   |            29.48           |        21.28        |   |   |

---

### Official Review · Reviewer_ZUu3 · 2023-11-02

**Soundness:** 2 fair
**Presentation:** 4 excellent
**Contribution:** 2 fair
**Rating:** 6
**Confidence:** 4

**Summary:**

This paper mainly discussed a self supervised data augmentation, here is the summary:
Authors propose SASSL, a data augmentation technique based on Neural Style Transfer that
naturally preserves semantic information while varying stylistic information (Section 4).
In addition to that, authors have empirically demonstrated an improved downstream task performance by incorporating our
method into MoCo v2, without additional hyperparameter tuning. We report more than 2% top-1
accuracy improvement on the challenging ImageNet dataset (Section 5.1).
Authors claimed our augmentation method learns more robust representations by measuring their transfer learning capabilities on five diverse datasets. SASSL improves linear probing performance
by up to 3.75% and fine-tuning by up to 1% on out-of-distribution tasks (Section 5.2).
Authors observed that balancing stylization using representation blending and image interpolation performs best, and adding external style datasets for transfer can improve performance (Section 5.3).

In general, supervised learning have been demonstrated a possible way to make data augmentation, this paper just made some contribution to this point.

**Strengths:**

I think neural transfer is a spot point in this paper. In self supervised learning and especially when this topic comes across with the field of data augmentation, it is necessary to raise way to capture the distribution of high dimensional data.

**Weaknesses:**

The downstream task is only about classification. Maybe it can happen to be successful.

**Questions:**

Can you or did you do some other downstream task to demonstrate the success of data augmentation (e.g. object detection?)

---

> ### Author Response · Authors · 2023-11-21
> **Response to Reviewer ZUu3**
>
> We thank the reviewer for appreciating our paper’s presentation and for the insightful feedback.
>
> >### 1. Additional downstream tasks
>
> We appreciate the reviewer's recommendation to evaluate our method on additional downstream tasks. In response, we conducted further experiments on few-shot learning. The results from one-shot and ten-shot experiments on ImageNet demonstrate that the representations learned using SASSL enhance performance in both scenarios. Please refer to our General Response (item 3) for the details.

---

### Official Review · Reviewer_PzRD · 2023-11-02

**Soundness:** 3 good
**Presentation:** 2 fair
**Contribution:** 2 fair
**Rating:** 3
**Confidence:** 5

**Summary:**

The paper proposes SASSL (Style Augmentations for Self-Supervised Learning) to enhance the data augmentation strategies in SSL. SASSL applies neural style transfer to input images and take the stylized images as the augmented samples for SSL training. Experiments on SSL and transfer learning have been conducted to evaluate the proposed approach. SASSL outperforms original MoCo v2 by 2% on top-1 classification accuracy on the ImageNet dataset.

**Strengths:**

1. Experiments on both SSL and transfer learning have been conducted to evaluate the proposed method. The method performs better than the original MoCO v2 on ImageNet SSL.

2. The proposed is easy to understand.

**Weaknesses:**

1. The paper shows limited technical novelties. It adopts the commonly used NST method CSN for SSL data augmentation, where both CSN and SimCLR have been widely used in existing literature. Although the authors claimed that this is the first work to adopt NST for SSL, I assume this method of a simple combination of two well-known methods does not bring much inspiration to the community. In addition, similar ideas have been explored in early literature [r1, r2].

2. The proposed idea (NST for SSL data augmentation) is only evaluated on CSN of NST and SimCLR of SSL. More different trials, such as adaptive instance normalization (AdaIN) of NST and more SSL frameworks should be evaluated to demonstrate the effectiveness of this idea. Furthermore, why would you call the normalization method as CSN? This method was noted as Conditional Instance Normalization (CIN) in original paper, and this notion CIN has been widely used in current NST literature.

3. Table 2 shows that not all style datasets can outperform the original MoCo V2 for Retinopathy and DTD. The optimal style dataset also varies across different target datasets. The choice of style dataset would be a problem.

4. Table 3 shows that the proposed method would even harm the performance when interpolation weight $\beta=1$ that represents a full exploit of stylized image. A manual selection of $\alpha$ and $\beta$ are required, for instance $\alpha, \beta \in [0.1, 0.3]$ in this paper. It's not clear how the authors determined this range. This range may also have to be customized for different datasets, limiting the practical usage of the method.

5. It lacks insights into why NST works well for SSL data augmentation. Figures 2 and 3 do not provide sufficiently informative clues. They could probably be put into the appendix. The authors are encouraged to show more evidence of improvements (other than the performance improvement) brought by NST for SSL tasks. For instance in my mind, may we visualize the neuron activations before and after NST to investigate the effect of NST on SSL backbones?

6. The notations in this paper are somewhat confusing. 1) $\beta^{(k)}$ denotes the trainable function in conditional style normalization, while $\beta$ denotes the interpolation factor of content and style images in Eq. 11. The authors should reconsider the notations for either of them; 2) Should $f$ be $h$ in Eq. 2? 3) In Eq. 7 $\mathcal{T}$ is a stylization network, while in Eq. 1 $\mathcal{T}$ denotes all possible augmentations.

[r1] Zheng, Xu, et al. "Stada: Style transfer as data augmentation." arXiv preprint arXiv:1909.01056 (2019).

[r2] Jackson, Philip TG, et al. "Style augmentation: data augmentation via style randomization." CVPR workshops. Vol. 6. 2019.

**Questions:**

None

---

> ### Author Response · Authors · 2023-11-21
> **Response to Reviewer PzRD (1/2)**
>
> We thank the reviewer for the helpful comments and feedback.
>
> >### 1. Comparison to NST work on supervised learning
>
> We appreciate the reviewer bringing these recent work to our attention. In what follows, we will address their key differences from our proposed method.
>
> In their work [r2], Jackson et al. propose employing NST as a data augmentation technique for supervised learning, where the style reference is drawn from a normal distribution. However, our experiments demonstrate that this approach hinders performance in the context of SSL (as shown in item 4). Instead, SASSL utilizes precomputed style representations from external datasets or samples from the content datasets themselves (in-batch stylization), resulting in improved or on-par downstream performance across style datasets.
>
> Furthermore, their approach solely considers the blending of style representations, overlooking the distortions introduced by the stylization network itself. Our work reveals that the compression generated by the stylization network, even without using a style representation, leads to a loss of information about the structure (edges and small image details) of pre-training images, deteriorating downstream performance. This is the rationale behind SASSL's reliance on both pixel interpolation and feature blending. While both techniques employ NST as a content-preserving transformation, SASSL enables its use by incorporating it into the default data augmentation pipeline of various SSL methods (as shown in our General Response, item 2), which differs from utilizing NST for domain transfer tasks in supervised learning.
>
> As mentioned in Section 2.2, Zheng et al. [r1] utilize NST to create additional samples for supervised learning, using only eight style images. It's important to note that their approach does not incorporate NST as data augmentation; instead, the stylized images are fixed and added to the original training dataset. Additionally, their experiments are limited to a single classification dataset (Caltech). This contrasts sharply with our method, which proposes different stylization strategies (external and in-batch) and ways to control the stylization effect (via ablation and interpolation) to augment data on-the-fly in order to enhance SSL performance. While both methods share the use of style NST, their tasks and approaches are distinct.
>
> >### 2. Details on normalization
>
> In Section 3.2, we use the name Conditional Style Normalization to follow Fast Style Transfer’s official implementation. Please note that the [official codebase]((https://github.com/magenta/magenta/blob/main/magenta/models/image_stylization/ops.py)) makes a distinction between Conditional Instance Normalization, Weighted Instance Normalization, and Conditional Style Normalization. Following the reviewer’s suggestion, we have renamed the normalization method to Conditional Instance Normalization in our paper to prevent any confusion with the NST literature.
>
> >### 3. Style dataset selection
>
> When working with a target dataset where style/texture is not meaningful for classification (e.g., ImageNet and iNat21), all the evaluated style datasets enhance downstream performance. This means that **selecting any of these style datasets leads to better performance than using the baseline**. However, for target datasets where semantic information is intertwined with stylistic attributes, employing NST as data augmentation does not guarantee improved performance.
>
> Retinopathy and DTD datasets embed semantic information within their textures and local patterns. Retinopathy images employ local abnormalities and textures to determine the stage of diabetes. Similarly, DTD relies on patterns and appearance to distinguish between texture categories. In both cases, the semantic information is intrinsic to the texture itself. As a result, SASSL produces image representations that achieve performance comparable to the baseline (within one standard deviation).
>
> To provide a clearer understanding of these scenarios, we have extended our discussion and included a visual analysis that illustrates the differences between styles in different datasets (See Appendix Sec. A.2 for more details).

---

> ### Author Response · Authors · 2023-11-21
> **Response to Reviewer PzRD (2/2)**
>
> >### 4. Effect of feature matching
>
> Due to computational constraints, we are unable to perform additional experiments on alternative NST methods with different feature matching techniques. Instead of focusing on how features are matched using alternative alignment techniques (e.g. Whitening and coloring transforms, histogram matching, optimal transport, adaptive instance normalization, etc.), we provide a more in-depth examination of how the feature matching itself affects downstream performance. We conducted additional ablation studies to gain a better understanding of the impact of feature alignment, as described next.
>
> We ran additional experiments to analyze the downstream task performance of models pre-trained via MoCo v2 under four different scenarios: (i) default augmentations, (ii) SASSL with no conditional instance normalization (no feature alignment, only image encoding), (iii) SASSL using Gaussian noise instead of style representations for conditional instance normalization, and (iv) SASSL using external style representations from Painter by Numbers.
>
> The following table shows that the performance boost obtained via NST relies on both the image encoding and feature alignment processes, where feature alignment has a greater impact on performance. Without aligning the content representation to those of a style reference, performance improves by 0.8%. Aligning content representations using noise as a style reference decreases performance compared to using no alignment. Finally, using an external dataset as style representation improves performance by an additional 1.61%. This suggests that feature alignment is an important part of the stylization process, but it is not the only factor. The encoding component of NST, which is common across stylization methods, also contributes to improved representation learning (See Appendix Sec. A.7 for more details).
>
> | Augmentation Settings |              Stylization Mode              | Top-1. Acc (%) | Top-5 Acc. (%) |   |
> |:---------------------:|:------------------------------------------:|:--------------:|:--------------:|---|
> |        Default        |                      -                     |      72.97     |      90.86     |   |
> |         SASSL (Ours)        | Image encoding only (no feature alignment) |      73.77     |      91.64     |   |
> |        SASSL (Ours)         |   Gaussian noise as style representation  |      73.21     |      91.17     |   |
> |         **SASSL (Ours)**         |    **External dataset (Painter by Numbers)**   |      **75.38**     |      **92.21**     |   |
>
> >### 5. Hyperparameter selection
>
> The intervals for alpha and beta (0.1 to 0.3 in both cases) were selected by sweeping their minimum and maximum values and measuring their downstream performance on ImageNet. Similar to the other augmentations in MoCo v2, e.g., random cropping, color jittering and solarization, we fixed these intervals and used them in all our transfer-learning experiments.
>
> We respectfully disagree with the reviewer's assessment of the practical limitations of our proposed method. While further performance gains might be possible with different alpha and beta intervals, our experiments show that our method consistently enhances performance across datasets without the need for hyperparameter tuning. This aligns with most augmentations in MoCo v2 and other SSL methods, which rely on multiple hyperparameters yet still improve representation learning with fixed hyperparameter values. Our NST augmentation functions similarly and can be seamlessly integrated into the default augmentation pipeline to further enhance downstream performance.
>
> >### 6. Details on ablation study (Table 3)
>
> Table 3 demonstrates the significance of combining the intermediate stylized and content images using an interpolation factor beta. While preserving the entire intermediate stylized image negatively impacts performance, it is important to note that this is not our final proposed technique but rather the scenario where the interpolation step is disabled. As a result, fine-grained content details are lost due to the bottleneck compression introduced by the stylization network.
>
> Several NST methods require blending and interpolation because feature alignment techniques tend to overemphasize style attributes (as seen in the Universal Style Transfer implementation [r3]). So, it is expected that fully utilizing the stylized image hinders performance. The performance drop caused by not having control over the pixel interpolation is comparable to that observed in other SSL methods like BYOL, where removing a single augmentation can lead to a decline in top-1 accuracy of over 8% (refer to BYOL's paper, Appendix F.3).
>
> [r3] Li, Yijun, et al. "Universal style transfer via feature transforms." Advances in neural information processing systems 30 (2017).
>
> >### 7. Math notation
>
> We thank the reviewer for the suggested changes. We have updated the mathematical notation on the paper accordingly.

---

### Author Response · Authors · 2023-11-21
**General Response (1/2)**

We thank the reviewers for their valuable feedback, and we have made changes to the paper to address the issues raised, as detailed below. We also feel that there may have been some misunderstanding regarding the paper’s contributions and significance, which we have attempted to rectify with further revisions. We kindly request the reviewers to consider our response with an open mind. We look forward to further discussion and addressing any additional questions.

We begin by clarifying our contributions and answering some common questions about performance on different methods and tasks. Individual questions are answered separately to each reviewer.

>### 1. Contribution and Novelty

We propose a new self-supervised representation learning (SSL) technique called SASSL, which utilizes neural style transfer (NST) as a data augmentation method. SASSL separates content and style attributes from pre-training images, enabling us to modify the appearance of images in terms of texture and color schemes while preserving the semantic information encoded in the content attributes. We empirically show that SASSL achieves superior performance across multiple datasets, as evidenced by both linear probing and fine-tuning evaluations. We briefly summarize our findings:
- We show that using NST as data augmentation enhances SSL performance across various methods, including MoCo v2 (Table 1), SimCLR, and BYOL (see item 2).
- We propose a method to seamlessly integrate NST into standard SSL data augmentation pipelines by applying it to a single view during pre-training, controlling the stylization strength in both the latent space (blending) and pixel domain (interpolation), and utilizing both external and in-batch stylization (Section 5.1). We have found this configuration to be effective across various pre-training strategies.
- We have empirically shown that styles derived from the pre-training dataset itself (in-batch stylization) and external standard datasets (pre-computed style latent vectors) are sufficiently diverse to generate robust representations, improving downstream performance (Section 5.2).
- We have validated the hypothesis that object-centric datasets (where semantics are encoded in the content attributes) are invariant to NST, allowing style transformations to generate more effective encodings in a self-supervised manner (Section 5.3).

In addition, to demonstrate the transfer learning capabilities of our proposed method and provide a method for selecting a style dataset, we have included in our revised manuscript a visualization analysis based on the similarities between style latent vectors from different datasets. By representing style vectors using t-SNE embeddings, we show that image representations learned using SASSL improve transfer learning when the pre-training, style, and target datasets have a significant overlap in the t-SNE space (see Appendix Section A.2).

>### 2. SASSL performance on SimCLR and BYOL

As suggested, we ran additional experiments covering different SSL methods and the effect of SASSL in the downstream performance. To evaluate the generalizability of SASSL beyond MoCo v2, we conducted these additional experiments with SimCLR and BYOL, two other representative SSL methods. We pre-trained a ResNet-50 backbone using each method with and without SASSL, and then linearly probed the resulting models on ImageNet.

In both cases, we use their default pre-training and linear probing settings. SASSL was implemented with its reported hyperparameters (blending and interpolation factor between 0.1 and 0.3, Painter by Numbers used as external style dataset).

The results, summarized in the following table, show that SASSL consistently improves the downstream classification accuracy of both SimCLR and BYOL, suggesting its effectiveness across different frameworks (See Appendix Sec. A.5 for more details).

| Augmentation Settings |            Stylization Mode           | Top-1. Acc (%) | Top-5 Acc. (%) |   |
|:---------------------:|:-------------------------------------:|:--------------:|:--------------:|---|
|   SimCLR (Baseline)   |                   -                   |      68.62     |      88.7      |   |
| **SimCLR + SASSL (Ours)** | External dataset (Painter by Numbers) |      **69.58**     |      **89.01**     |   |
|    BYOL (Baseline)    |                   -                   |      74.09     |      91.83     |   |
|  **BYOL + SASSL (Ours)**  | External dataset (Painter by Numbers) |      **75.13**     |      **92.12**     |   |

---

> ### Author Response · Authors · 2023-11-21
> **General Response (2/2)**
>
> >### 3. Additional Tasks
>
> To further demonstrate the enhanced representation learning capabilities of SASSL, we conducted additional experiments on another downstream task: few-shot classification. Due to computational constraints, semantic segmentation results using SASSL are not yet available.
>
> We evaluated the MoCo v2 representations obtained through SASSL against the default MoCo v2 model in the context of few-shot learning, specifically on ImageNet's one-shot and ten-shot settings.
>
> The table below presents the classification accuracy in both scenarios. The results reveal that SASSL boosts few-shot classification accuracy by over 1% in terms of top-1 accuracy in both cases. This aligns with our classification experiments, suggesting that SASSL promotes more general image representations (See Appendix Sec. 6 for more details).
>
> | Augmentation Settings |            Stylization Mode           | Top-1. Acc (%) One-shot | Top-1 Acc. (%) Ten-shot |   |
> |:---------------------:|:-------------------------------------:|:-----------------------:|:-----------------------:|---|
> |        Default        |                   -                   |          19.56          |          45.05          |   |
> |         **SASSL (Ours)**        | External dataset (Painter by Numbers) |          **20.55**          |          **46.73**          |   |

---

### Meta-Review · Area_Chair_LYgH · 2023-12-12

**Metareview:**

This submission received diverging reviews. One reviewer suggested 3 while others suggested 6. The rebuttal addressed the concerns of one of the reviewers while the other did not respond. Overall, the babe looks like a borderline so cannot be accepted due to some weaknesses reported by the reviewers. They include: lack of sufficient ablation study, lack of a detailed discussion about relevant data augmentation, lack of sufficient discussion of the related work.

**Justification For Why Not Higher Score:**

Some weaknesses include: lack of sufficient ablation study, lack of a detailed discussion about relevant data augmentation, lack of sufficient discussion of the related work.

**Justification For Why Not Lower Score:**

N/A

---

### Decision · Program_Chairs · 2024-01-16

Reject